# The Electric-Field-Driven Fusion Jetting 3D Printing for Fabricating High Resolution Polylactic Acid/Multi-Walled Carbon Nanotube Composite Micro-Scale Structures

**DOI:** 10.3390/mi11121132

**Published:** 2020-12-21

**Authors:** Xiaoqiang Li, Guangming Zhang, Wenhai Li, Zun Yu, Kun Yang, Hongbo Lan

**Affiliations:** Shandong Engineering Research Center for Additive Manufacturing, Qingdao University of Technology, Qingdao 266520, China; li17852429258@163.com (X.L.); l1664941485@163.com (W.L.); zunyu168@163.com (Z.Y.); yangkunqut@163.com (K.Y.); hblan99@126.com (H.L.)

**Keywords:** electric-field-driven, polymer matrix composites, 3D printing

## Abstract

Existing 3D printing techniques are still facing the challenge of low resolution for fabricating polymer matrix composites, inhibiting the wide engineering applications for the biomedical engineering (biomimetic scaffolds), micro fuel cells, and micro-electronics. In order to achieve high resolution fabrication of polylactic acid (PLA)/multi-walled carbon nanotube (MWCNT) composites, this paper presents an electric-field-driven (EFD) fusion jetting 3D printing method by combining the mixing effect and material feeding of the micro-screw and the necking effect of Taylor cone by the EFD. The effects of main process parameters (the carbon loading, the voltage, the screw speed, and the printing speed) on the line width and the printing quality were studied and optimized. To demonstrate the printing capability of this proposed method, meshes with line width of 30 µm to 100 μm and 1 wt.% to 5 wt.% MWCNT for the application of conductive biomimetic scaffold and the anisotropic flexible meshes were prepared. The electrical properties were investigated to present the frequency dependence of the alternating current conductivity and the dielectric loss (tanδ), and the microstructures of printed structures demonstrated the uniformly dispersed MWCNT in PLA matrix. Therefore, it provides a new solution to fabricate micro-scale structures of composite materials, especially the 3D conductive biomimetic scaffolds.

## 1. Introduction

Polymer matrix composites (PMCs) have been widely used in the fields of aerospace, automotive, biomedical, flexible sensors, and tissue engineering (tissue stents and nerve catheters) [1,2,3,4,5] due to their excellent mechanical properties, thermal properties, and the electrical properties (electrical conductivity and electromagnetic shielding) [6,7,8]. Especially the PMCs with the addition of various nanomaterials, such as one-dimensional nanomaterials (carbon nanotubes, nanowires, and nanofibers), and two-dimensional nanomaterials (such as graphene), possess more excellent comprehensive performances [6,9,10]. However, the fabrication of PMCs mainly relies on the traditional processes such as the compression molding, fiber placement (winding) molding, injection molding, and casting molding. These traditional PMCs molding processes have the difficulties in achieving the complex 3D structures and high-resolution structures, which severely limits their engineering applications.

The 3D printing technology has provided a new solution for the manufacturing of PMCs. In recent years, researchers have proposed a variety of molding processes based on 3D printing, including fused deposition modeling (FDM) [11,12], direct ink writing (DIW) [13], digital light processing (DLP) and stereo lithography (SLA) [14,15,16], selected laser sintering (SLS) [17,18], and the electrospinning [19,20]. Among them, the FDM has been regarded as one of the most popular ways because of its low manufacturing cost and good process universality. For examples, Heinrich et al. [5] prepared polycarbonate (PC)/multi-walled carbon nanotube (MWCNT) sensors with line width of 600~800 μm. Gnanasekaran et al. [11] fabricated PBT/carbon nanotube (CNT) samples with different loadings (0.5 wt.%, 1 wt.%, 2 wt.%, 4 wt.%) and the line width of 300–500 μm. Huang et al. [21] prepared polycaprolactone (PCL)/MWCNT scaffolds with loading (0.25 wt.%, 0.75 wt.%, and 3 wt.%) and the line width of 330 μm for adipose stem cell culture. However, due to the limitations of nozzle and molding characteristics of FDM, it is very hard for FDM to achieve high resolution fabrication (below 100 μm) of PMCs [22].

In order to achieve high resolution fabrication of polylactic acid (PLA)/multi-walled carbon nanotube (MWCNT) composites, we present a new process of electric-field-driven (EFD) fusion jetting 3D printing method, which combines the mixing effect and material feeding of micro-screw and the Taylor cone necking effect of the EFD fusion jetting to achieve high-resolution printing of PMCs structures. Firstly, the working principals of this method has been introduced. Subsequently, the influences of process parameters such as voltage, screw speed, printing speed, and the loading of MWCNT on the line width and printing quality were studied. Lastly, in order to prove the printing capability of this proposed method, several typical structures have been printed, and their electrical properties and microstructures with different loadings of MWCNT were investigated.

## 2. Materials and Methods

### 2.1. Experimental Platform and Working Principals

Figure 1 shows the electric-field-driven fusion jetting 3D printing system, in order to achieve the fabrication of polymer matrix composites, this 3D printing system is designed with a printhead, a movement module, a high-voltage power, a signal generator, and a target substrate. Among them, the target substrate is mounted on the movement module, which is programmed to move in X-, Y- with a stage resolution of 1 µm. The printhead is composed of a micro-screw, stepping motor, cylinder, heater, and the conductive nozzle. The micro-screw driven by a stepping motor is designed with a diameter of 15 mm, a helix angle of 17°, and the total length of 200 mm for material feeding, and the conductive nozzle is connected to the high-voltage supply to generate electric field. In addition, a polyetheretherketone (PEEK) part (Figure 1d) was used to separate the cylinder and the conductive nozzle in case of the short-circuit is this system due to the indispensable high-voltage.

Then, the printing mechanism can be described as follows (seen in Figure 1c): the pre-mixed composite powders are melted by the heater and further mixed by shearing and extrusion effect of the micro-screw. Thus, the melted composite is extruded to the tip of the nozzle, where it forms a meniscus under surface tension. When the high positive potential voltage is applied, the high electric potential of the nozzle will cause the redistribution of charges in the substrate by the effect of electrostatic induction. Negative charges accumulate on the top surface of substrate, while the positive charges move to the bottom surface. Then, an induced electric field will be generated between the nozzle and the substrate. The pendent meniscus at the tip of the nozzle formed by the air pressure will be affected by the induced electric field. Under the coupling effects of the electric field force, surface tension, and viscous force, the pendent meniscus will be elongated gradually to form a Taylor-cone. A fine jet from the apex of the cone will be produced and deposited on the substrate with micro-scale/sub-micro-scale line structure only if the tangential electric stress exceeds the surface tension. This process will be continued to print layer by layer for forming a desired composite 3D structure.

### 2.2. Raw Materials for 3D Printing

The thermoplastic material of polylactic acid (PLA, nature works PLA-4032D powder) with powders size of 58 μm was purchased from Dongguan Dansheng Plastic Co. LTD (Guangdong, China). The MWCNT powders with fiber length of 3–12 µm purchased from the composite materials E-buy Technology Co. LTD (Guangdong, China) have been used as the reinforcement. According to the designed loading of nanomaterials, the PLA powders with different loadings of MWCNT powders were ball milled for 3 h at a speed of 200 rpm for pre-mixing. Then, the pre-mixed composite powders were dried at 80 °C at least 2 h in a drying oven.

### 2.3. Evaluation of Microstructure and Performance

To evaluate the performance of 3D printed PLA/MWCNT composites, several measurements were conducted. The AC conductivity and the dielectric loss (tanδ) were measured with the frequency range of 10^−1^–10^6^ Hz by the wideband dielectric spectrum tester (Concept 80). Microstructures of the printed specimens with different loading were observed with SEM (MERLIN Compact-62-24) and optical microscope (OLYMPUS DSX-510). The conductivity and resistance were measured with the high-power DC power supply (MESTEK DP3020). Each sample was measured three times to obtain the average value and error bar.

## 3. Results

### 3.1. The Optimization of Process Parameters

In the 3D printing process of PLA/MWCNT composites by the EFD fusion jetting 3D printing method, there are many process factors including the screw speed (feeding flow), printing speed, voltage, and the loading of nanomaterials which may have influences on feature size and the printing quality of the produced PLA/MWCNT composites. Therefore, these parameters were investigated in the present research. The parameters are shown in Table 1.

Figure 2 shows the influences of process parameters (the voltage, the printing speed, and the screw speed) on line width and printing quality of two composite materials with different loading of MWCNT. On the whole, these two composite materials present an almost similar trend of line width with the three process parameters. As shown in Figure 2a, with an increase in the voltage from 1000 V to 1550 V, the line width firstly increases to a peak, then slightly decreases and keeps stable finally. It can be explained that an increasing voltage will cause larger amount of polarized charges at the tip of the meniscus of melt composite materials, leading a larger electric field force to form a thicker jet [23]. When the voltage is larger, the amounts of polarized charges reach a maximum, causing slightly decreased line width to stable terrace. One should note that the voltage for maximum line width of 5 wt.% MWCNT/PLA is 1350 v, while that of 2 wt.% MWCNT/PLA is 1250 V. The line width (range of 88–145 μm) of composite with 5 wt.% MWCNT is always thicker that (range of 30–74 μm) of 2 wt.% MWCNT of composite. It indicates that the increasing loading of MWCNT will induce more charges at the tip of meniscus to generate a larger electric field force [24]. Meanwhile, the too low voltage cannot maintain a continuous printing due to the insufficient induced charges (shown in the inset of Figure 2a). Therefore, the voltage of 1050 v was chosen for next investigation.

Additionally, during the EFD fusion jetting 3D printing, the line width also depends on the printing speed and the screw speed. Figure 2b shows the effect of printing speed on the line width, which decreases continuously with the increase of printing speed. This is mainly ascribed that the Taylor cone jet are stretched to be a thinner line due to the effect of the increasing viscous drag force inside the material with the movement of the printing platform by the increasing printing speed. Contrary to the printing speed, as the screw speed increases, the line width always increases linearly, shown in Figure 2c. Considering of round shape of line (diameter of R), the jet flow (volume of printed line of πR^2^/4) increases in the speed of the squared line width. It means that the increasing jet flow rate relies on not only the linearly increasing of the materials supply with increasing rotate speed of screw, but also the decreasing viscosity due to the increasing shear rate of screw.

Meanwhile, the printing speed and screw speed should match each other for getting a good printing quality. As shown in Figure 2d–f, there are three samples printed with different printing speed and screw speed. When the printing speed (35 mm/s) is too high comparing with the screw speed (6 rpm), the feeding is insufficient. Thus, the printed line is broken, and the surface of the line is relatively rough, as shown in Figure 2d. When the printing speed (25 mm/s) is too low comparing with the screw speed (6 rpm), the materials supply will be overfeeding, resulting in uneven printing lines and large drop points, as shown in Figure 2f. The print quality of Figure 2e printed with the printing speed of 30 mm/s and the screw speed of 6 rpm is better than Figure 2d,f. Only if a certain matching relationship between printing speed and screw speed is built, the surface of the printed sample will be smooth, as shown in Figure 2e. Therefore, considering the small line width (20–60 μm for 2 wt.% and 80–120 μm for 5 wt.%) and good printing quality, the parameters of the screw speed (4–6 rpm), and the printing speed (30–35 mm/s) were chosen for next investigation.

### 3.2. The Fabrication and Evaluation of PLA/MWCNT Composites

At present, conductive 3D biomimetic scaffolds have shown great potentials in electrical stimulation-assisted spinal cord injury [21,25,26]. However, the current preparation methods of conductive biomimetic scaffolds face challenges of very few attachments due to the large line width and large pitch. For example, Huang et al. [21] used FDM technology to prepare PCL/MWCNT scaffolds with the line width 330 μm using the different loading of CNT (0.25 wt.%, 0.75 wt.%, and 3 wt.%) for adipose stem cell culture. Lee et al. [25] prepared 0.1 wt.% MWCNT-hydrogel composite biomimetic scaffolds with a line width of about 200 μm by SLA technology for nerve cell culture. Ho et al. [26] prepared 400–800 μm PCL/MWCNT scaffolds with CNT loading of 5 wt.% by DIW technology for H9c2 cells culture. In order to demonstrate the capability of EFD fusion jetting 3D printing, several micro-scale structures such as the single mesh and multi-layer mesh have been printed. By adjusting the above process parameters (shown in Table 2) and using a nozzle with an inner diameter of 300 μm, the single-layer meshes with the MWCNT loading of 1 wt.% and the line width of 30 μm (shown in Figure 3a,b), and the MWCNT loading of 2 wt.% and the line width of 60 μm (shown in Figure 3c,d) were printed. At the same time, in order to fabricate a conductive 3D biomimetic scaffold with a much smaller line width and a higher MWCNT loading than those printed by FDM, a six layers mesh with 5 wt.% MWCNT and the line width of 100 μm was prepared successfully, which can be connected with a 32 v power supply to lit a small lamp.

Zigzag structures with the MWCNT loadings of 2 wt.% and 5 wt.% (shown in Figure 4a–d) were also printed. Because of the difference in the structures of orthogonal elements, these structures can be bent and stretched in the diagonal direction, while can hardly be stretched in the other directions (shown in Figure 4b,d). Figure 4e presents the conductivity and resistance with different MWCNT loadings. As increase of the MWCNT loading, the conductive increased from 0.17 ± 0.05 S·m^−1^ to 0.6 ± 0.05 S·m^−1^ and the resistance decreased from 5.76 ± 0.3 Ω·m to 1.66 ± 0.3 Ω·m. One should note that the structure with low MWCNT loading is hardly to measure conductivity due to the too thin line width (less than 300 μm). The zigzag structures with MWCNT loading of 2 wt.% were connected with a 32 v power supply to lit a small lamp (shown in Figure 4f), which proves the possibility of EFD fusion jetting 3D printing method on fabricating the conductive micro-structures.

Furthermore, in order to further study the conductivity of PLA/MWCNT composites for the applications of conductive 3D biomimetic scaffolds, especially, to explore electrical conductivity while conducting an AC stimulus in the biological experiment, the electrical properties of composite materials with different contents were tested. Figure 5a presents the frequency dependence of the complex electrical conductivity of the real part with different amounts of MWCNT. It is readily seen that the trace of 2 wt.% is typical of disordered materials with values of AC conductivity increasing with frequency, varying slowly first and then more rapidly at high frequencies; this variation follows a power law exponent with a value very close to 1. The highest AC conductivity values obtained here of 2 wt.% and 5 wt.% for the highest frequency of 10^6^ Hz are 10^−8^ and 10^−6^ S/cm, respectively. These values are much smaller than the reported results of 10^−5^ S/cm for 2 wt.% CNT/PCL and 10^−4^ S/cm for 4 wt.% CNT/PCL [27], which may be induced by the small line width and small area of line junction of these printed micro-scale structures by the EFD fusion jetting 3D printing. Figure 5b show the frequency dependence of dielectric loss (tanδ) of 2 wt.% and 5 wt.% MWCNT/PLA composite materials. The dielectric loss (tanδ) increases with the decrease of frequency, showing significant interfacial polarization characteristics, indicating that a large number of microscopic interfaces were introduced into the composites by the MWCNT filler. Figure 5c,d show the SEM images of the distribution of 2 wt.% and 5 wt.% MWCNT in the composite materials. It can be seen that the MWCNT distributed evenly in the PLA matrix. As the MWCNT loading increases, the denser the dispersion of MWCNT in the PLA matrix. Thus, it is easier for the MWCNT to overlap each other, which improves conductivity of composite materials. Therefore, the printed meshed structures with different loading and different line width have proved that the equipment can realize high resolution fabrication of the PLA/MWCNT composites.

## 4. Discussion

This paper presents an EFD fusion jetting 3D printing method by combining the mixing effect and material feeding by the micro-screw and the Taylor cone necking effect of the EFD fusion jetting for fabricating micro-scale structures of PLA/MWCNT composites. Through the experimental investigations, the following conclusions can be obtained:(1)The influences of processing parameters (the carbon loading, voltage, screw speed, and printing speed) on the line width and printing quality of printed parts were studied. The line width increases with the increase of carbon loading and screw speed, firstly increases to a peak and then keeps stable with the increase of voltage, and decreases with the increase of printing speed. Considering both the small line width (20–60 μm for 2 wt.% and 80–120 μm for 5 wt.%) and good printing quality, the parameters have been optimized as the screw speed (4–6 rpm), the voltage of 1050 v, and the printing speed (30–35 mm/s).(2)Two single-layer meshes with line width of 30 µm and 60 µm, a six-layer conductive biomimetic scaffold with 5 wt.% MWCNT and the line width of 100 μm for the application of conductive biomimetic scaffold, and the anisotropic flexible conductive meshes have been prepared successfully for demonstrating its printing capability.(3)For the composites with loading range of MWCNT from 2 wt.% to 5 wt.%, the AC conductivity increases, and dielectric loss (tanδ) decreases with the increasing of frequency. The highest AC conductivity values obtained here of 2 wt.% and 5 wt.% for the highest frequency of 10^6^ Hz are 10^−8^ and 10^−6^ S/cm, respectively, and the SEM images shows the uniformly distributed MWCNT in the PLA matrix of composites with both 2 wt.% and 5 wt.% MWCNT.

In conclusion, the proposed technique can provide a new and industrially applicable solution for high resolution PMCs fabrication. In the future work, we will further apply it to print the other composite materials such as the graphene and carbon fiber, and to fabricate the 3D biomimetic scaffolds for cell culture in vitro and animal experiment.

## Figures and Tables

**Figure 1 micromachines-11-01132-f001:**
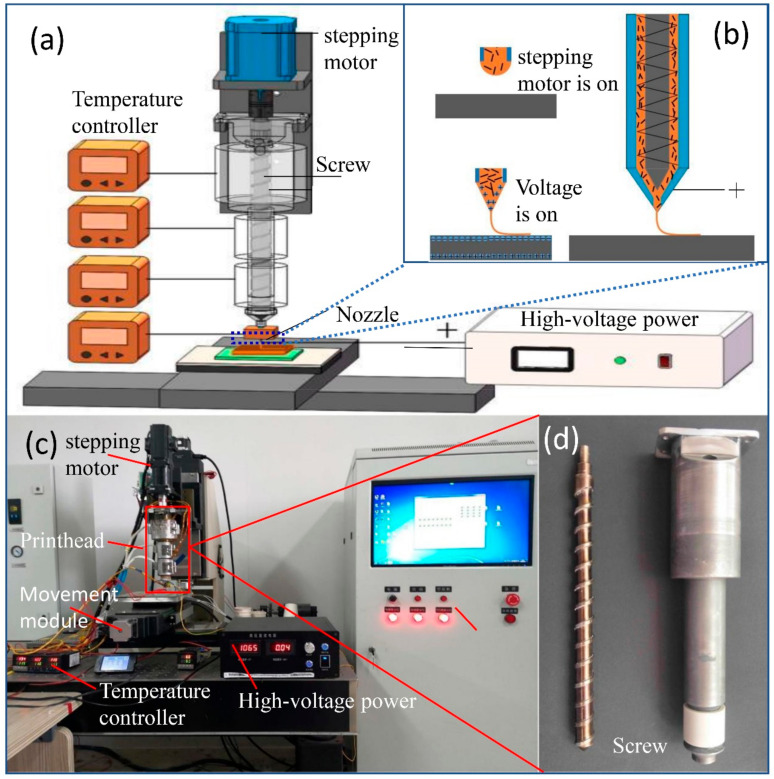
The electric-field-driven fusion jetting 3D printing for polymer matrix composites: (**a**) the schematic of the 3D printing system; (**b**) The printing mechanism; (**c**) the 3D printer; (**d**) the micro-screw.

**Figure 2 micromachines-11-01132-f002:**
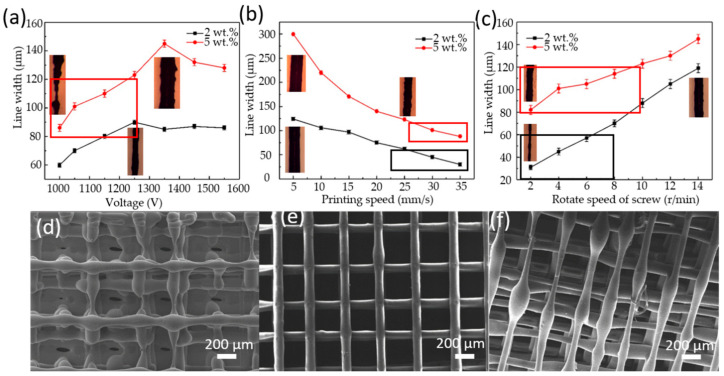
Influences of process parameters on the line width: (**a**) voltage; (**b**) printing speed; (**c**) screw speed, in which the red box and black box are the acceptable parameters for the 2 wt.% and 5 wt.%, respectively; The printed results of polylactic acid (PLA)/multi-walled carbon nanotube (MWCNT) composite with loading of 5 wt.% in the condition of: (**d**) underfeeding; (**e**) proper feeding; (**f**) overfeeding.

**Figure 3 micromachines-11-01132-f003:**
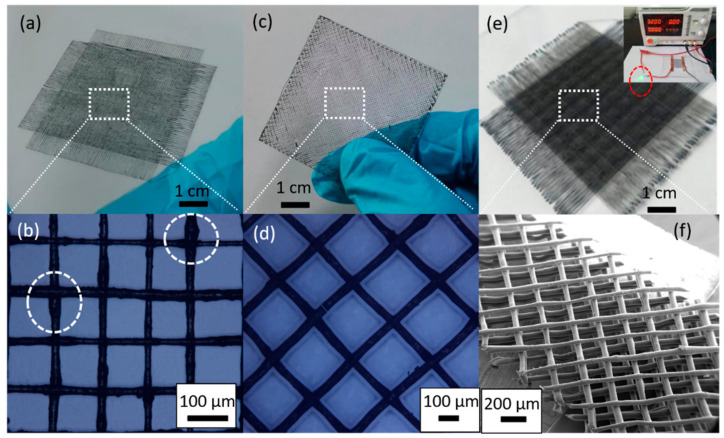
Micro-scale structures of the PLA/MWCNT composite: (**a**,**b**) the single layer mesh with line width of 30 µm and loading of 1 wt.%; (**c**,**d**) the single layer mesh with line width of 60 µm and loading of 2 wt.%; (**e**,**f**) the 6 layers mesh with line width of 100 µm and loading of 5 wt.%;.

**Figure 4 micromachines-11-01132-f004:**
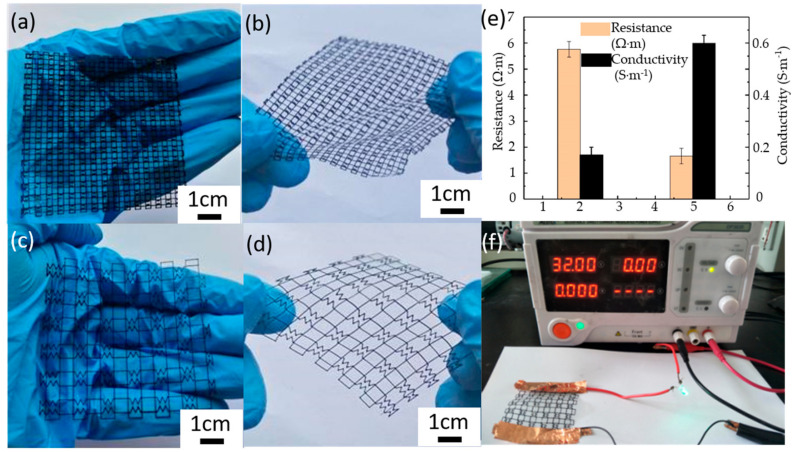
The anisotropic flexible mesh of the PLA/MWCNT composite: (**a**,**b**) the loading of 2 wt.%; (**c**,**d**) the loading of 5 wt.%; (**e**) the conductivity and resistance; (**f**) the conductive experiment of flexible mesh.

**Figure 5 micromachines-11-01132-f005:**
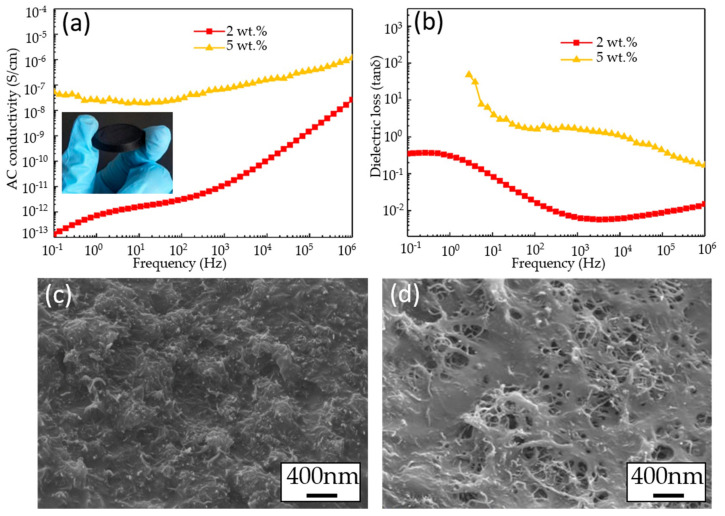
(**a**) The AC conductivity and (**b**) the dielectric loss (tanδ) of PLA/MWCNT composites with the loading of 2 wt.% and 5 wt.%; The microstructure of PLA/MWCNT composites with the loadings: (**c**) 2 wt.%, (**d**) 5 wt.%.

**Table 1 micromachines-11-01132-t001:** Investigating parameter of EFD jetting 3D printing process.

Target Parameter	Range	Other Parameters	Other Parameters
Voltage (U/v)	1000, 1050, 1150, 1250, 1350, 1450, 1550	V_s_: 4 rpm; V_p_: 30 mm/s	Inner diameter of nozzle: 300 μm; Standoff height: 400 μm; Loading of MWCNT: 2 wt.%, 5 wt.% Temperature: 140
Screw speed (V_s_/rpm)	2, 4, 6, 8, 10, 12, 14	U: 1050 V; Vp: 30 mm/s	
Printing speed (V_p_/mm/s)	5, 10, 15, 20, 25, 30, 35	U: 1050 V; f7Vs: 4 rpm	

**Table 2 micromachines-11-01132-t002:** The processing parameters for printing different line width.

The Samples	Screw Speed	Printing Speed	Voltage	Printing Temperature	Standoff Height
Figure 3a,b 30 μm	4 rpm	35 mm/s	1050 V	140 °C	400 μm
Figure 3c,d 60 μm	6 rpm	30 mm/s	1050 V	140 °C	
Figure 3e,f 100 μm	4 rpm	30 mm/s	1050 V	140 °C	

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
