# Peer review of "The Electric-Field-Driven Fusion Jetting 3D Printing for Fabricating High Resolution Polylactic Acid/Multi-Walled Carbon Nanotube Composite Micro-Scale Structures"

_micromachines, 2020, doi:10.3390/mi11121132_

Round 1

Reviewer 1 Report

The topic of the manuscript entitled ‘The electric-field-driven fusion jetting 3D printing for fabricating PLA/MWCNT composite micro-scale structures’ falls within the scope of the journal Micromachines. The paper contains interesting experimental results. However, the manuscript requires major corrections.

Comments and remarks are presented below.

  1. Authors should explain the academic contribution of the work developed. Highlighting what is innovative/original about the existing literature.
  2. In the Introduction section, authors should describe the structure of the paper.
  3. The abbreviations used in the article should be explained in detail the first time use them (eg. page 1, line 17 - PLA/MWCNT).
  4. Figure 2 should be reorganized or the descriptions in the figure enlarged - figures (a -c) are illegible.
  5. Caption under Figure 2 is on the next page.
  6. The caption under Figure 5 should be redrafted.
  7. The manuscript should contain a chapter Conclusions.
  8. Conclusions should be presented in points.
  9. In the chapter Conclusions, the authors should also indicate the directions of further research.
  10. References are made unworthily with the Instructions for authors and template. In addition, references account for over 20% of the entire content of the article. This is not a review article.
  11. The authors of the article should carefully check and correct it so that it was consistent with the Instructions for authors and template.

Reviewer 2 Report

The present paper by Li and co-authors deals with 3D printing of PLA/MWCNT composites for applications such as conductive biomimetic scaffolds.

They study the influence of several parameters such as the screw speed, printing speed, voltage and loading of nanotubes on the final material.

They first investigate the line width as a function of the different varying parameters. The influence of each parameter is studied separately and trends on the line width are given as a function of the varying parameter. It is however not clear how the authors define the target line width, i.e., they should add on graphs 2a-c for instance the range of acceptable line width which would allow us understand the range of input parameters that can be employed for printing this composite.

I would expect a more thorough analysis of the  parameter influence on the printing quality.  The authors should provide a clear map of the acceptable parameters leading to acceptable printed solutions (indicating clearly what they expect as acceptable first).

They should also motivate the choice of the final parameters to print the final scaffolds, based on the recommendation they will provided from the analysis described before.

The figures lack quality, and the axis on graph are not visible. They should be put in the same fontsize as the text for the sake of clarity.

Please make the analysis on the influence of the %wt of MWCNT clearer. Please clearly indicate which parameter vary in the analysis. It seems that not only the %wt of MWCNT varies but also the width of the printed line.

A more thorough investigation of the microstructures should be provided. The link between the microstructure and the scaffold properties should be clearly established.

It is worth noting that the paper is rather well written, but the authors should go further in their analysis, giving more thorough discussions and presentations of each points. For these reasons, I recommend major revisions of the paper before resubmitting.

Please make a great effort on figures!!! The fontsize must be the same as in the text, and good quality figures must be provided, otherwise the paper cannot be published with such low quality figures.

Miscellaneous:

Please define the MWCNT (I guess multi-wall carbon nanotubes) acronym. It is actually defined in the text but should be explicated in the paper title.

Please improve the quality of fig 1, especially concerning the text within the figure.

Please improve figure 2. The graph axis are barely visible, use the same fontsize as in the text.

L125 - much large, => larger

Please indicate in Fig 2 d-f the parameters corresponding to under/overfeeding or having a final good print quality

L172: Figure ? what figure?

L173: Figure ? what figure?

Figure 4: the axis of the inset (e) are not visible. Please take care to provide visible figures with sufficient quality

Actually, the inset in Fig 4 (e ) is already given in the text. Please remove this inset and add the uncertainty intervals in the text when mentioning the mean values. How these uncertainties were quantified? It must be clearly stated.

Quality of Fig 5.a,b Is not acceptable. Please provide good quality figures.

Round 2

Reviewer 1 Report

In fact, Figure 5 should be divided into two figures, e.g .:

Fig. 5. Electrical properties of PLA / MWCNT composites: (a) AC conductivity, (b) dielectric loss (tanδ).

Fig. 6. Microstructure of PLA / MWCNT composites with loadings: (a) wt 2%, (b) wt 5%.

Moreover, in figure captions may be omitted articles (words): the, a, an.

However, these are minor deficiencies that do not affect the quality and substantive value of the article.

The authors took into account all comments of the reviewer and in my opinion the paper can be published in its current form.

Reviewer 2 Report

Ok for publication.